# Indirect Effects of Ebola Virus Disease Epidemics on Health Systems in the Democratic Republic of the Congo, Guinea, Sierra Leone and Liberia: A Scoping Review Supplemented with Expert Interviews

**DOI:** 10.3390/ijerph192013113

**Published:** 2022-10-12

**Authors:** Philippe Mulenga-Cilundika, Joel Ekofo, Chrispin Kabanga, Bart Criel, Wim Van Damme, Faustin Chenge

**Affiliations:** 1Centre de Connaissances en Santé en République Démocratique du Congo, Kinshasa 3088, Democratic Republic of the Congo; 2School of Public Health, Faculty of Medicine, University of Lubumbashi, Lubumbashi 1825, Democratic Republic of the Congo; 3Institute of Tropical Medicine, 2000 Antwerp, Belgium

**Keywords:** Ebola, health service delivery, health workforce, health financing, health information system, leadership, governance

## Abstract

Ebola Virus Disease (EVD) epidemics have been extensively documented and have received large scientific and public attention since 1976. Until July 2022, 16 countries worldwide had reported at least one case of EVD, resulting in 43 epidemics. Most of the epidemics occurred in the Democratic Republic of Congo (DRC) but the largest epidemic occurred from 2014–2016 in Guinea, Sierra Leone and Liberia in West Africa. The indirect effects of EVD epidemics on these countries’ health systems, i.e., the consequences beyond infected patients and deaths immediately related to EVD, can be significant. The objective of this review was to map and measure the indirect effects of the EVD epidemics on the health systems of DRC, Guinea, Sierra Leone and Liberia and, from thereon, draw lessons for strengthening their resilience vis-à-vis future EVD outbreaks and other similar health emergencies. A scoping review of published articles from the PubMed database and gray literature was conducted. It was supplemented by interviews with experts. Eighty-six articles were included in this review. The results were structured based on WHO’s six building blocks of a health system. During the EVD outbreaks, several healthcare services and activities were disrupted. A significant decline in indicators of curative care utilization, immunization levels and disease control activities was noticeable. Shortages of health personnel, poor health data management, insufficient funding and shortages of essential drugs characterized the epidemics that occurred in the above-mentioned countries. The public health authorities had virtually lost their leadership in the management of an EVD response. Governance was characterized by the development of a range of new initiatives to ensure adequate response. The results of this review highlight the need for countries to invest in and strengthen their health systems, through the continuous reinforcement of the building blocks, even if there is no imminent risk of an epidemic.

## 1. Introduction

Ebola virus disease (EVD) is a serious and often fatal disease caused by the Ebola virus (EBOV) [1]. This virus belongs to the genus *Ebolavirus of the Filoviridae* family of single-stranded negative SENSE RNA viruses [2]. The first documented outbreak of EVD was reported in 1976 in the village of Yambuku in the north of the Democratic Republic of Congo (DRC) [3].

The EBOV is primarily transmitted by human-to-human contact through infected body fluids and cadavers [4]. The incubation period is usually 5–9 days, with a range of 1–21 days, and patients are not considered infectious until they develop symptoms [5]. In the early stage of infection, patients present with a nonspecific febrile illness (symptoms may include lack of appetite, arthralgia, headache, malaise, myalgia and a rash) that progresses over the first week to severe gastrointestinal symptoms and signs (nausea, vomiting and diarrhoea) [1]. As the Ebola virus load increases, the severity of the clinical manifestations of EVD also increases. The onset of detectable viremia and clinical signs and symptoms in most patients occurs 6–10 days after exposure [1].

Diagnosis requires a combination of clinical case definition and laboratory tests, usually real-time reverse transcription PCR to detect viral RNA or rapid diagnostic tests based on immunoassays to detect EBOV antigens [1].

A randomized controlled trial identified two effective antibody-based therapies for EVD, but much more needs to be done to improve outcomes [6]. Combination therapies, including antivirals combined with a high level of supportive care, could improve outcomes in severely ill patients with the highest risk of death [6].

Vaccines to protect against Ebola have been developed and have been used to help control the spread of the recent EVD outbreaks in Guinea and in DRC [7]. However, the current situation with Ebola vaccination shows the need for increased community engagement in public health interventions [8].

Since 1976, up until July 2022, WHO [7] counts 16 countries worldwide that have already reported at least one case of EVD, resulting in 43 epidemics. Most epidemics occurred in DRC (14 outbreaks, with a total of 4754 cases and 3222 deaths) but the largest epidemic occurred from 2014–2016 in Guinea, Sierra Leone and Liberia in West Africa (28,610 cases, 11,308 deaths). Twenty-two epidemics occurred in other African countries and totalled 1452 cases and 851 deaths. A total of 7 cases have been exported outside Africa (Appendix A).

In most affected African countries, health systems are failing and responses to the epidemic are often (too) slow; the health systems lack health professionals skilled in screening, supportive treatment, contact tracing and surveillance [9]. In addition, the health systems are underfunded, with out-of-pocket payments as the main source of health financing. There is an insufficient investment in the implementation of health policies based on primary health care, whereas more attention would have enabled these countries to build more resilient health systems.

In such a context, the indirect effects of an EVD epidemic on a health system can be significant [10,11]. Indirect effects are the consequences beyond infected patients and deaths immediately related to EVD. The literature also indicates that studies on the indirect effects provide important information but few of them have been carried out on the scale of all affected areas. Evidence does indeed indicate that EVD epidemics have further weakened the health systems of the affected countries [10]. Given that most epidemics occurred in DRC, and the largest one in Guinea, Sierra Leone and Liberia, this study mostly focused on the indirect effects of the EVD epidemics in those countries.

The objective of this study was to map and measure the indirect effects of the EVD epidemics on the health systems of DRC, Guinea, Sierra Leone and Liberia and, from thereon, draw lessons for strengthening their resilience vis-à-vis future EVD outbreaks and other similar health emergencies.

## 2. Materials and Methods

This scoping review, supplemented with expert interviews, followed the methodological guidelines described by Arksey and O’Malley [12] in 2005 and by Levac, Colquhoun and O’Brien [13] in 2010. We then (1) identified the research question; (2) conducted a literature search; (3) selected the studies; (4) extracted the data; and (5) summarized and reported the results.

We also drew on the PRISMA checklist “Extension Scoping Review (PRISMA-ScR)" [14] to align this scoping review with the “gold standard” method in this field.

The research question we defined was “What are the indirect effects of EVD epidemics on the health systems in the Democratic Republic of the Congo (DRC), Guinea, Sierra Leone and Liberia?”

To determine the indirect effects of EVD epidemics on health systems, we used the WHO frameworks on epidemic management [15] and health systems [16], which assess the effects of epidemics on the six building blocks of a health system [17]. The six building blocks are: (1) service delivery, which involves the use and organization of medical resources, equipment, prevention and other patient-centred services; (2) the health workforce, in which a “high-performing” country has a responsive and productive supply of trained health workers who are readily available at all times; (3) information, which includes the development of health information, surveillance systems and standardized tools and instruments and the collection and publication of international health statistics; (4) financing, which involves raising sufficient funds both internally and through external sources to ensure that people have access to necessary health services and are protected from catastrophic health expenditures or impoverishment; (5) medical products, vaccines and technologies, and their procurement and use to protect populations from health disparities; and (6) leadership and governance, which includes governments forming coalitions, working with external actors and developing policies to help the health system protect citizens [16].

For this sixth building block of the health system, we also used the governance analysis framework of Mikkelsen-Lopez I et al. [18] to further identify indirect effects.

The sub-questions that arose from the six building blocks were formulated as follows: “What are the indirect effects of EVD epidemics on (1) the service delivery; (2) the health workforce; (3) the health information system; (4) the health financing; (5) the essential medicines; and (6) the governance and leadership? in the DRC, Guinea, Sierra Leone and Liberia?”

We conducted a literature search in PubMed and in the gray literature to identify documents in English or French (articles, reports) that were relevant to the study of indirect effects of EVD on health systems and published between 1976 (the year of the first documented epidemic of EVD which led to the discovery of the EBOV) [3] and 2020. The search strategy used terms and medical subject heading keywords (MeSH) are as follows:Ebola virus diseaseHealth service delivery in AfricaHealth workforce in AfricaHealth information systems in AfricaHealth care financing in AfricaEssential medicines in AfricaLeadership or governance in Africa[(1) and (2)] or [(1) and (3)] or [(1) and (4)] or [(1) and (5)] or [(1) and (6)] or [(1) and (7)]

In this review, we included documents that met the following criteria: (1) published in English or French; (2) regarding EVD epidemics and containing one or more indirect effects of these epidemics on the health system in DRC, Guinea, Sierra Leone and Liberia; (3) published between 1976 and 2020; and (4) with an available abstract and full text that was freely accessible. The exclusion criteria used to filter the different documents were as follows: (1) documents published in a language other than French or English; (2) documents without abstracts or freely available text; (3) documents regarding EVD epidemics outside of Africa; and (4) documents dealing with direct sickness and death associated with EVD. We did not restrict the search to a particular study design or type of article. We also checked the references of all selected studies to identify additional articles that met our inclusion criteria.

The literature search was conducted by three researchers (PM, JE, CK). They first checked each article using the title and abstract to discern those that met the inclusion criteria. For all abstracts that met the inclusion criteria, full-text articles were analysed. Finally, the research team extracted data relating only to the indirect effects of EVD. The relevant information was recorded on a grid that contained headings related to each building block of the health system (Appendix A). The diagram of the literature search strategy is presented in Figure 1.

As there may be indirect effects of EVD epidemics on health systems dimensions not covered in published papers, we supplemented this review with interviews with 11 experts in the field [19]. The selected experts were closely involved in EVD outbreak preparedness and/or response in DRC or in West Africa (Guinea-Sierra Leone-Liberia). During the epidemics, they were working for ministries of health, bi- or multilateral agencies (CDC, WHO) or for NGOs (MSF) (Table 1).

A topic guide for the interviews was developed and pretested. Interviews were conducted online by three of the authors (PM-C, JE and CK). Each interview lasted on average 30 min and was recorded on a digital recorder with prior approval of the participants. The results were manually transcribed by each interviewer and then extracted into a Microsoft Excel table.

Quotes from interviews with the experts were identified by the code INT followed by a two-digit number, such as INT XX (Table 1).

## 3. Results

Of the 86 documents analysed in this scoping review, most were original articles (62%) and reviews (10%). All other documents (reports, supplements, viewpoints, commentaries, communication, critiques, debates, editorials, informational, public notes, perspectives, briefs) accounted for 28%. Almost all of the documents included in this study (98.8%) were published in 2014 and after.

It should also be noted that most of these papers were about the Guinea, Sierra Leone and Liberia 2014–2016 epidemic (*n* = 63, 73%). The DRC, which is the country with the most outbreaks of EVD, was covered by only seven papers included. Some of the papers included in the review dealt with more than one building block of the health system (Table 2).

In the following paragraphs, we report on the documented indirect effects of EVD epidemics on the health system building blocks in the four targeted countries.

### 3.1. Delivery of (Essential) Health Services

Disruptions in the supply and use of essential health services (outpatient consultation, hospitalization of patients, delivery in the presence of qualified personnel, treatment of confirmed cases of malaria and vaccinations) during EVD epidemics were well documented [9,20,21,22,23,24,25,26,27,28,29,30,31,32,33,34,35,36,37,38,39,40,41,42,43,44,45,46,47,48,49,50,51,52,53,54,55]. These disruptions were due to health workers’ fears of contracting EVD [39], community distrust of health workers and limited availability of health services in health care facilities [46] or the closure of some health facilities [20,21,22,23,24,25,26,27,28,29,30,31,32,33,34,35,51,52,53,55,56,57,58,59,60,61,62]. According to the experts, the closure of state health care facilities led the population to discover alternative treatment routes with many possible consequences. One of the interviewed experts stated that *"It’s the private sector that provides much more care to the general population*” [*INT 09*]. More financial inaccessibility characterized the affected populations, as one expert stated in the following quotes: “*Populations in areas affected by EVD outbreaks are further impoverished as their economic activities are severely impacted*; *… in the private health structures, it is clear that the cost of care became a major obstacle for the majority of the local population*” [*INT 02*]; another expert stated that “*people preferred to go to traditional practitioners*” [*INT 11*].

A general decrease in curative consultations was noted [20,28,29,53,54,55,60,63]. Ansumana et al. observed a decrease of up to 50% in outpatient consultations in health facilities in Guinea, Sierra Leone and Liberia [20]. An expert in the DRC stated that “*people consulted the structures less, especially state structures where you have the implementation of the intervention*” [*INT 09*].

Deliveries in health facilities also decreased [9,20,22,25,28,30,31,32,33,34,35,36,37,38,39,53,55,63,64,65,66]. This decrease was at least 50% in West Africa during the 2013–2014 epidemic [20]. It was estimated to be more than 60% in Guinea and more than 80% in Sierra Leone during the 2014–2015 epidemic [33,36].

Many authors also reported significant disruptions in immunization activities during EVD epidemics [20,24,27,31,38,40,51,52,53,54,55,64,67,68]. For example, in Liberia, Penta-3 vaccination coverage decreased by 26%, from 76% in 2013 to 50% in 2014 [20,40]; in addition, measles coverage decreased from 74% in 2013 to 58% in 2014 [40].

Several studies reported a reduction in coverage of antenatal and postnatal consultations [20,22,23,28,29,30,34,35,36,38,44,52,53,64,65]. In Sierra Leone, the literature noted that the number of women attending a fourth antenatal visit during the EVD epidemic fell by 27% [22]. Liberia had a 9–14% reduction in antenatal visits from the peak number before the EVD epidemic [22].

Many other studies noted a significant decrease with sometimes dramatic consequences in the use and capacity of care for certain disease control programs (tuberculosis, malaria and HIV/AIDS) [9,20,22,23,25,26,27,28,29,38,41,42,50,51,52,53,54,55,60,63,68,69,70,71]. Indeed, tuberculosis caused approximately 11,900 deaths in Guinea, Liberia and Sierra Leone in 2014. Of the TB deaths, approximately 2164 (95% CI 1815–2548) in Sierra Leone, 3463 (95% CI 2808–4349) in Guinea and 2164 (95% CI 1815- 2548) in Liberia, and were estimated to have increased by the EVD epidemics. [20]. With regards to malaria, there was a decline in malaria consultations (e.g., a 60.5% decline in malaria cases diagnosed and treated in health facilities in Sierra Leone) during the EVD epidemics [9,20,22,23,25,27,28,29,38,41,42,50,60,63,69]. For HIV/AIDS, there were drug shortages that led to the deterioration in the quality of care, an increase of more than 40% of treatment dropouts in Guinea and Liberia [25] and a decrease in the use of HIV/AIDS services [22,23,25,36,38,52,69].

Admissions, both in basic and emergency departments, as well as surgical interventions in hospitals were also affected [21,22,31,36,55,64,68]. In Sierra Leone, during the 2014–2015 EVD epidemic, the authors noted a 41% decrease in weekly surgical interventions [21].

Declines in the use of family planning services were reported during EVD epidemics [28,36,38,44,55]. During the 2014–2015 epidemic, this decline was estimated at 51% in Guinea [36], while in Sierra Leone, an increase of more than 1.2 million unintended pregnancies was reported in 2015 [28].

To some extent, the various EVD epidemics also positively influenced health service delivery. This was noticeable in the following: the establishment of several new community health centres [48]; the involvement or engagement of the community in addressing public health risks in Guinea [26]; the intensification of community engagement and dialogue with traditional and religious leaders in the measles vaccine campaign in Guinea, Liberia and Sierra Leone during the 2014–2015 epidemic [27]; improvements in the conditions of port health services and personal hygiene [31,72]; investments in community-based health service delivery in Liberia [32]; the creation of a national rapid response team to support committees during major public health events in Liberia [46]; the arrival of NGOs to support the response and health facilities in Sierra Leone [39]; strengthening of the health care system in general and improving occupational safety in medical settings in at-risk countries [73]; increases in household hygiene practices; changes in social interaction between affected people and the community and promoting hand hygiene practices [50,74]; strengthening the capacity of existing laboratories to perform microbiological and serological tests for all common diseases [75]; and the provision of free health care in the health zones affected by EVD in Equateur/the DRC [76].

### 3.2. Health Workers

On the one hand, health workers were severely affected during the EVD epidemics. In particular, the reduction in and even absence of the workforce during these epidemics in the four studied countries was noted [9,20,24,26,27,29,30,32,33,35,37,42,49,51,65,73,77,78,79,80,81]. This was either due to health workers leaving health facilities because of the fear of contracting and dying of EVD while providing services [9,24,32,35,37,64,65,77,79]. This has further reinforced the shortage of health care providers in the world, which Ngatu et al. estimated at 7.2 million [73]. However, the opposite was observed in the DRC, i.e., the (temporary) influx of health workers to health facilities supported the response. One expert stated that “*On the other hand, what I observed was that people were coming because there was money in Ebola*” [*INT 01*]. This situation disrupted health services, as another expert stated in the following quote: “*We have seen people leaving the unaffected health areas to go to the affected areas, in order to make money*; *this must have disrupted their duty stations too*” [*INT 03*].

Health workers in the facilities complained of an increased workload and occupational stress [9,22,39]. In the communities, health workers were subject to stigma, blame and social exclusion [77].

On the other hand, the EVD epidemics provided opportunities for capacity building for health care providers (clinicians, nurses, midwives, etc.) [41,49,70,73,78,79,81,82,83,84,85,86,87] and training for community health workers and local leaders [40,82]. This capacity-building focused, for example, on surveillance (including community-based surveillance), field epidemiology, infection prevention and control practices, mental health (stigma and stress management), burial practices and the use of paper health records (PHRs) [48,77,78,81,82,84,85,87].

To cope with the increased workload, additional health workers were recruited and deployed to areas affected by the EVD epidemics [9,26,35,54,88,89,90]. Guinea, for example, recruited 2950 workers during the 2014–2015 epidemic [26].

### 3.3. Health Information System (HIS)

During the EVD epidemics, some studies reported problems in the management of the health information system [27,53,54,55,62,63,91]. These problems included the malfunctioning of the surveillance systems. These systems were characterized by disruptions of monthly data collection and analysis [9,27,53,62,63,91]. The dysfunction in surveillance was also due to the disregard of existing tools by government partners. One expert stated that *"each partner came with their own software to experiment with and the government did not have a hand in evaluating, sorting and selecting what was relevant and what would help the Ministry of Health and the government. Knowing that we had our DIHS2 software that should be valued*; *no, each partner came with their software, and this situation was not advantageous for the country*” [*INT 02*].

Positive indirect effects in terms of several initiatives to support or strengthen the HISs as a whole were taken during the EVD epidemics [70,83,92,93,94,95,96,97]. For example, an expert in the DRC even stated that “*there was the development of the ‘DHIS2 tracker’, which can be used during emergencies to track contacts, carry out analyses in a short time, produce reports that are of great value and share them at the time to allow decision-makers to provide or propose solutions to the situation, and the creation of ‘Emergency Operations Centres in Kinshasa, Equateur, Beni, and Butembo*” [*INT 02*].

The EVD epidemics prompted the need for global disease surveillance and monitoring, better international preparedness for similar epidemics, and meet the needs of the press by providing reliable information [98]; additionally, the activation of the existing national polio surveillance system was prompted in Nigeria [99].

### 3.4. Financing

The mobilization of financial resources to better manage epidemics was of concern to policy-makers. In this review, this mobilization was considered to be a positive indirect effect of EVD on the financing building block. These effects included the reallocation of funds from other less urgent health initiatives to the EVD response [25,53,70,72,100] and the influx of funds for strengthening the health system [9]. A call for an international health systems fund to respond to public health emergencies that imperil the routine functioning of health systems, such as the 2014–2015 EVD outbreak in West Africa, was triggered [99]. In the DRC, for example, one expert said that “*during the Likati outbreak in Lokolia, there was no ferry to cross the Lomela River. Almost 2 to 10 years ago, the population crossed by pirogues. However, the intervention team needed to arrive in Lokolia*; *it was in August 2014 that we should cross. However, it was thanks to Ebola funds that we were able to repair the ferry and that helped to open the health zone*” [*INT 02*].

The Ebola funds that were left over after the epidemics were used to manage other epidemics in the DRC, as this expert’s quote indicates: *‘Out of all the funds that were intended for the response to the 10th epidemic, there were funds that remained, which were diverted to the fight against cholera in Kasai’* [*INT 10*].

The diversion of funds to the response from other activities created other problems in the management of other health programs. One expert stated that “*However, there were also investments, for example, we had problems with the lack of Early Detection Tests for malaria, while the money instead went to the acquisition of PCI (Prevention Control and Infection) inputs for Ebola*” [*INT 06*].

### 3.5. Essential Medicines, Medical Products and Equipment

The EVD epidemics negatively impacted the supply chains of essential medicines and medical products. Affected countries experienced stock-outs of malaria tests and medicines [23,42,52,54,55]. The supply of medicines was insufficient to cover all the facilities, and these medicines were soon out of date because they were not used in the less-frequented health facilities. In the DRC, two experts stressed that “There were *drugs that we received from partners, but which were not sufficient for all the facilities*” [*INT 01*]; “*However, it must be said that many of them had expired because of the use of services for fear of being diagnosed as Ebola-positive*” [*INT 03*]. The decrease in the supply of oral antimalarials was very noticeable, but there was an increase in the use of injectable antimalarials [23].

The EVD epidemics prompted actions to assist the affected populations, not only for Ebola-virus-diseased patients or control activities but they could also be used for other patients with other problems not related to EVD. These actions included the provision of additional ambulances and vans for health committees [48], the availability of means of transport such as motorbikes and ambulances for health facilities to transport patients [30,70] and the provision of new drugs in the various clinics [101], and the scaling up of the health system for the safe transport of samples, supply of reagents and disposal of hazardous materials. The equipment or materials were also intended to be used for other health emergencies as declared by this expert: “*there were motorcycles, bicycles, computers, vehicles and generators which were provided to health structures in order to strengthen their automobile transport, their logistical capacities and prepare for new health emergencies*”.

### 3.6. Leadership and Governance

This review identified indirect effects related to three of the governance dimensions described by Mikkelsen-Lopez et al. [18]: long-term strategic vision and policy design, the participation of all relevant stakeholders and accountability.

The interviewed experts clearly identified problems in the management of EVD epidemics. These included the creation of ad hoc EVD response structures in which national and local health authorities were not fully involved, as one expert said: *‘The Health Zone Manager was not involved in the management of Ebola in his health zone. Even the Provincial Health Division was not involved*”; “*It was partners leading the response and the system*; *the ministry no longer had the leadership*” [*INT 01*].

The onset of EVD prompted policy-makers to launch initiatives that either allowed for the proper management of epidemics in general or for the support of affected patients [9,24,30,36,39,46,72,75,77,83,84,85,86,87,96,99,102,103,104]. Examples to illustrate this include: the creation of health crisis or emergency management bodies, advisory committees or national rapid response teams, as well as mental health and psychosocial support services for EVD patients. One expert even stated that “*Ebola response coordination bodies have been replicated to manage other health emergencies*” [*INT 02*]. The experts stated that there was systematically an emphasis on multisectoral action. In the field, however, they experienced a different reality. One expert pointed out: “*The emphasis is on the multisectoral nature of the responders. However, in the field, this was not the case*; *it was more the medicalization of the response. In addition, it was only after a lot of insistence that you would see experts from other sectors following suit*” [*INT 02*].

In the DRC, to promote access to health facilities during outbreaks of EVD, the government initiated free health care in affected health areas [76].

## 4. Discussion

This is the first review mapping the indirect effects of EVD on health systems in the four most affected African countries.

Before discussing the actual results of the review, it is relevant to note that the relatively limited number of studies conducted in the DRC, which is the country that has historically been most frequently affected by EVD epidemics, is striking. This may suggest a reporting bias. Although the search strategy covered the long period from 1976 to 2022, virtually all articles included in this review were published from 2014 on. At that time the DRC was experiencing its 7th EVD epidemic, and the three West African countries under study were experiencing the largest health-system-devastating EVD epidemics with the occurrence of cases in the North [8]. In this health security context, the concept of health system resilience reappeared in the discourse of donors and policymakers. We believe that this led to an increased focus on health system research in the context of EVD epidemics, especially in the three West African countries. In this review, the information gap on DRC was, nevertheless, partly filled by interviews with experts.

The results of this study indicate that health service delivery was severely disrupted during EVD outbreaks. During outbreaks of EVD, activities such as immunization were interrupted, leading to outbreaks of other infectious diseases [20,24,27,31,38,40,64]. Health services for the management of diseases other than Ebola were almost non-existent during the epidemics [9].

Disruptions in the supply of health care (curative consultations, vaccination services, maternity services, etc.) observed during the outbreaks of EVD were comparable to those observed during major outbreaks of other infectious diseases, such as the severe acute respiratory syndrome (SARS) epidemic of 2003 in Asia [105]. SARS had a negative impact on the use of medical services in Taiwan. As with EVD, this viral disease led to significant reductions in outpatient care (23.9%) and inpatient care (35.2%) [105]. In both epidemics, people’s fear of contracting the disease appeared to have a strong impact on their access to care. Other authors also indicated that the COVID-19 pandemic had a profound impact on health care delivery systems [106,107]. We believe that this negative impact could have been mitigated by making the health system more resilient, in particular by strengthening community structures, strengthening the role of political, religious, cultural leaders and health workers in mobilizing and involving community members helps to dispel myths and perceptions related to outbreaks of infectious diseases in the community and to fight against slack. The involvement of the community in monitoring, care and follow-up could be a guarantee of success. In the DRC, for example, the COVID-19 pandemic led to significant reductions in the use of health services. The lifting of the lockdown led to a rebound in the utilization of health services, but nevertheless, remained below previous levels [107].

In view of the disruption of service provision, it is important to acknowledge that these devastating effects of EVD on the fragile health systems of the affected countries have caused more deaths than the virus itself, particularly among vulnerable populations, namely, pregnant women and children under five years of age [108]. For example, a significant rise was observed in comprehensive emergency obstetric and newborn care facilities (CEmONCs) across Sierra Leone for the maternal case fatality rate (incidence rate ratio (IRR): 1.44, 95% CI: 1.17, 1.75) and for stillbirths (IRR: 1.37, 95% CI: 1.16, 1.39) [22]. All affected countries had never before experienced outbreaks of EVD, but they have, repeatedly, had other deadly outbreaks, like cholera for instance, and have quite some experience with the management of those outbreaks. A major difference between those other outbreaks and the EVD epidemic is that the latter has generated much more international attention; one of the reasons being the fact that the very nature of the transmission of EVD is such that it can be “imported” elsewhere, and thus, also in the North, as was the case during the 2014–2016 EVD epidemic in West Africa; EVD cases were imported in Italy, Spain, the United Kingdom and the United States of America.

Due to poor working conditions and fear of contracting the disease, health workers decided to go to other countries when the epidemics started [9,24,30,48,65]. This situation led to a shortage of skilled labour, further limiting patients’ access to care [67]. To cope with this situation, decision-makers deployed other personnel (national or foreign) to ensure a continuity of care [9,26,35,88,89]. Similar situations have also been experienced in the current pandemic of COVID-19, where, in addition to physical and/or psychological exhaustion of health staff, Ministries of Health have encouraged retired medical staff and those who spend more time on research to resume their clinical work in order to try and overcome the increasing pressure on the health system [109]. Since controlling an epidemic requires skilled personnel, several initiatives were undertaken to strengthen or train staff [40,48,70,75,76,78,79,80,81,82,83,84]. Such initiatives were also taken in the COVID-19 pandemic. Many countries are forced to innovate to meet the needs of their populations, and the health workforce is at the very centre of this response [109].

Epidemic surveillance is problematic in countries affected by EVD [27,62,63,91]. Health workers lack the knowledge to control epidemics [8]. Tambo et al. point out that the key to monitoring and controlling epidemics is to have an effective and adequate surveillance response system with early warning and the ability to determine transmission projections [110].

As a result of weaknesses in the health systems of countries affected by EVD, governments and their partners have had to develop programmes to strengthen health data management [90,91,92,93]. It is important to strengthen surveillance, geographic information systems (GISs) and modelling to estimate disease projections [9].

In countries affected by EVD, there have been significant shortages of essential drugs such as antimalarials [23,42]. According to the WHO, a well-functioning health system ensures access to essential medical products [111]. Affected countries need to be prepared for each outbreak to avoid systemic disruption caused by shortages of essential medical supplies. Efforts made during previous outbreaks of other diseases in terms of improving patient transport and the supply of drugs or inputs for laboratory work should be supported [30,48,101,102]. Similar drug shortages were observed during the COVID-19 pandemic in Zimbabwe in 2020. In this context, the authors highlighted the disruption in prescription renewal schedules and routine laboratory surveillance tests for chronic pathologies [112].

EVD has prompted the governments of affected countries to work with national and international partners and research institutes in the creation of health crisis management bodies [9,24,46,77,83,102]. The vision was also to empower staff, in particular, to control epidemics.

Health governance and financing have a significant impact on the rest of the health system. Guinea, for example, experienced a collapse in funding for health activities because of EVD [26]. In most of the affected countries with weak health systems, investments were needed in health systems infrastructure for sustainable development. Funding must, therefore, include adequate payments, surveillance, the purchase of necessary supplies and research development [9].

The main limitation of this study is that we only used non-paying databases, which may have missed some relevant information. However, conducting interviews with key informants helped to fill this gap.

## 5. Conclusions

The results of this review showed that the EVD outbreaks in DRC, Guinea, Sierra Leone and Liberia had a negative impact on their health systems. To mitigate these effects, policymakers and health authorities from these countries need to set up and implement effective policies to continuously strengthen their health systems by addressing each of the six health system building blocks. By doing so, any health system will achieve a level of resilience that will enable it to respond promptly and effectively to future EVD outbreaks and other similar health emergencies. This mapping of the indirect effects of EVD on health systems is a preliminary study that can lead to further analyses, such as systematic reviews and meta-analyses.

## Figures and Tables

**Figure 1 ijerph-19-13113-f001:**
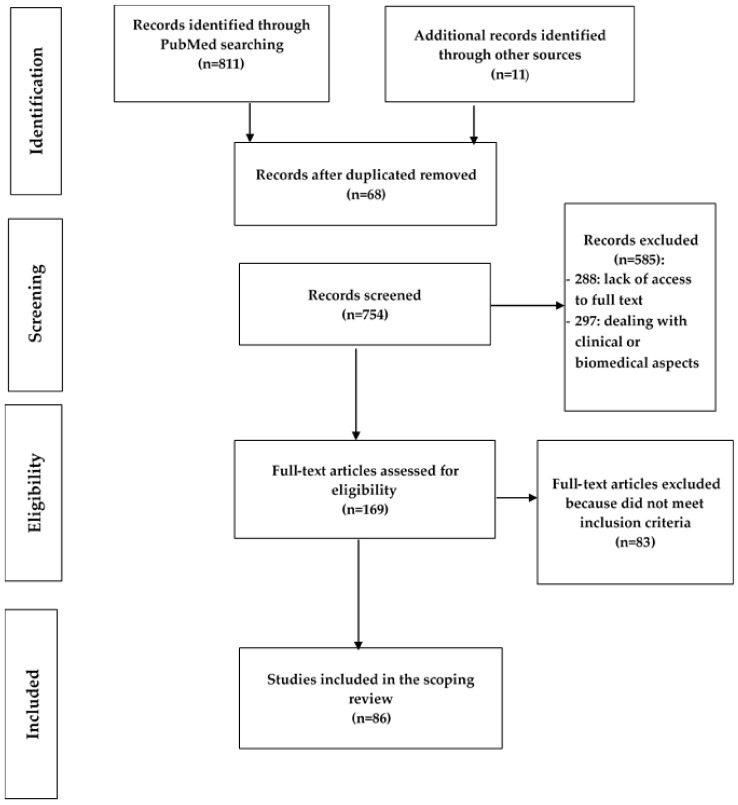
PRISMA diagram of the literature search strategy.

**Table 1 ijerph-19-13113-t001:** Details on the experts interviewed.

Expert Codes	Expert Affiliations	Expert Duty Country
INT 01	MoH	RD. Congo
INT 02	WHO/AFRO	RD. Congo and West Africa
INT 03	MoH	RD. Congo
INT 04	Africa CDC	RD. Congo and West Africa
INT 05	MoH	RD. Congo
INT 06	MoH	RD. Congo
INT 07	CDC	RD. Congo
INT 08	WHO/AFRO	RD. Congo and West Africa
INT 09	University	RD. Congo
INT 10	MSF	RD. Congo
INT 11	WHO/AFRO	RD. Congo

**Table 2 ijerph-19-13113-t002:** Nature of the health system building blocks addressed in the selected studies.

Health System Building Blocks	*N*	%
Service Delivery	59/86	68.6
Workforce	40/86	46.5
Health Information System	18/86	20.9
Financing	7/86	8.1
Medicines	9/86	10.4
Leadership and Governance	20/86	23.2

## Data Availability

The data that support the findings of this study are available from the corresponding author, Philippe Mulenga-Cilundika (drphilippe.mulenga@gmail.com), upon reasonable request.

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
