# Peer review of "Indirect Effects of Ebola Virus Disease Epidemics on Health Systems in the Democratic Republic of the Congo, Guinea, Sierra Leone and Liberia: A Scoping Review Supplemented with Expert Interviews"

_ijerph, 2022, doi:10.3390/ijerph192013113_

Round 1
Reviewer 1 Report
Estimated Authors,
I've read with great interest the present paper dealing with the indirect effects of Ebola Virus Outbreaks in several African countries. Through a scoping review model, Mulenga-Cilundika et al were able to stress several aspects that may appear counter-logical (e.g. the positive effects of Ebola virus outbreaks in terms of occupation rates for HCWs and infrastructural interventions that, in an otherwise low-resource settings were channeled in specific interventions for aiding the "Ebola fighters"). Interestingly, several of these aspects may find some similarities in the SARS-CoV-2 pandemic: if this paper had been published before the SARS-CoV-2 pandemic, several points that the authors were able to stress could have anticipated seveal critical issues, representing a true "guiding light" for Public Health Authorities.
From the point of view of the present reviewer, therefore, some further discussion about such similarities could radically improve the quality and the timely appreciation of this study.
From a formal point of view, unfortunately, the present reviewer think that some further improvements are required. More precisely:
1) Authors (by editing also Figure 1) should more precisely stress how many studies were ACTUALLY included in their study and why they were excluded; as several studies were not retrieved because of the unavailability of the full text, it should be more carefully addressed;
2) As the "question" is both interesting but quite vague (what is an indirect effect of an outbreak?), a PICO summary table should be implemented in the main text.
3) 86 studies are a relatively huge number of papers to be summarized, but at least as annex a summary table must be included allowing the readers to appreciate more carefully the main outcome and point stressed by parent study authors.
4) As there is a certain umbalance in reporting (e.g. DRC is interested in a relatively reduced number of paper, while this is the country that is historically more severely affected by the Ebola Virus), representing a substantial reporting bias, I would recommend either to separately report data for DRC vs. other countries or to more extensively discuss this issue.
Author Response
Dear Reviewer, I would like to refer you to the attachment
Regards,

Reviewer 2 Report
Mulenga et al. submitted a manuscript gathering data about disruption of healthcare services and activities caused EBOV outbreaks between 1976 and 2020. This is of interest for the journal and general public. This is well written but is not ready for publication in the present form.
I made some comments that need to be addressed as they would benefit the manuscript:
-EBOV Vaccine is available and licensed since 2019. It needs to be in the introduction. How come the authors are not mentioning it anywhere? while I agree that it directly relates to the virus it also involves non-virus related aspects (in the middle of an EBOV outbreak between 2019 and 2020) that should be discussed such as how well a new vaccine was accepted in Africa, the rate of vaccination, the potential logistic issues for delivery or challenge to get it where needed on time, the source or the lack of funding.
-Please pay attention to the nomenclature of Ebola virus. It is EBOV not EBV. Also, be consistent throughout the manuscript regarding how you use EBOV outbreak and EVD outbreak. Pick one. Line 43: add negative SENSE RNA viruses.
-Materials and Methods: the six building blocks are not listed in the same order between section 105-115 and section 119-122.
-The authors highlight an important limitation for their study: only free articles were used. Adding the exact number of studies that were excluded solely based on that criteria in the PRISMA diagram (out of 83) would help readers judge how much information was lost at that level.
-spell out DRC in abstract, remove after infection line 47, replace paper by manuscript or study line 81, remove "not on the ........ cases" line 82-83,
Author Response
Dear Reviewer, I would like to refer you to the attachment.
Regards,

Round 2
Reviewer 1 Report
The paper has been improved according to the previous recommendations, and no further reworking is requested.
Reviewer 2 Report
thank you for addressing my comments.